# Surface Plasmon Resonance Sensors Using Fractional Optical Vortices

**George Andrei Bulzan**[1, 2, ⋆]

**1** Physics Faculty, University of Bucharest, P.O. Box MG-11, 077125 Bucharest-Magurele, Romania;
**2** National Institute for Research and Development in Microtechnologies (IMT), Str. Erou Iancu Nicolae 126A, 077190 Voluntari, Romania

⋆ george.bulzan@imt.ro

## Abstract

This study focuses on the response on reflectance of a surface plasmon resonance (SPR) sensor when hit by a Laguerre-Gaussian (LG) beam with a fractional orbital angular momentum (OAM) quantum number. More exactly, it investigates the change of the position and value of its minimum point with respect to the situation in which plane waves without OAM were used. Thus, an analytical expression for this physical quantity has been obtained based on which numerical results have been plotted. Those results are similar to the case in which the OAM quantum number was an integer. However, the use fractional optical vortices (FOV) has numerous practical advantages regarding the manipulation of the molecules of the analyte.

## 1 Introduction

SPR sensors in Kretschmann configuration designed in figure 1 have multiple applications in multiple domains such as biomedicine [1], environmental safety [1] and combatting

bioterrorism [1]. This configuration consists on a prism put on top of a thin sandwich of metals that covers one of its surfaces [2]. All of this is put over an analyte whose index of refraction needs to be determined.

In one hand optical vortices (OV) present an helicoidal wavefront [3,4] and the number of twists in one wavelength is equal to its topological charge [3,4]. FOVs have by definition a fractional topological charge [5]. On the other hand FOVs present a slice in their annular shape [5] and have multiple applications in domains such as optical communication [6,7], optical imaging [8], quantum optics [9] and radar imaging [10].

Our paper will investigate the functionality of SPR sensors when illuminated with FOV. More exactly, we will calculate the dependance of the response on reflectivity on its OAM quantum number.

A similar study has been done for LG beams that have an integer OAM associated quantum number in [11]. Moreover, FOV can be interpreted as an infinite superposition of OV with integer OAM quantum numbers [12]. For this reason, they can be used for the same things the OV with integer topological charges can such as: pinning down large molecules at the surface [13], positioning them conveniently [13], optical trapping [14] or manipulating nanoparticles (despite the fact that FOVs make the rotation of the particle more difficult [15]). This means that FOVs and OVs with integer topological charges have the same practicability. However, FOVs can also realize cell sorting [5,15] and cell orientation control [5,16] which makes them useful if the analyte is i.e. a biological liquid. Thus, this study represents a continuation of the investigation done in [11] by comparing the reflectance of FOVs and plane waves.

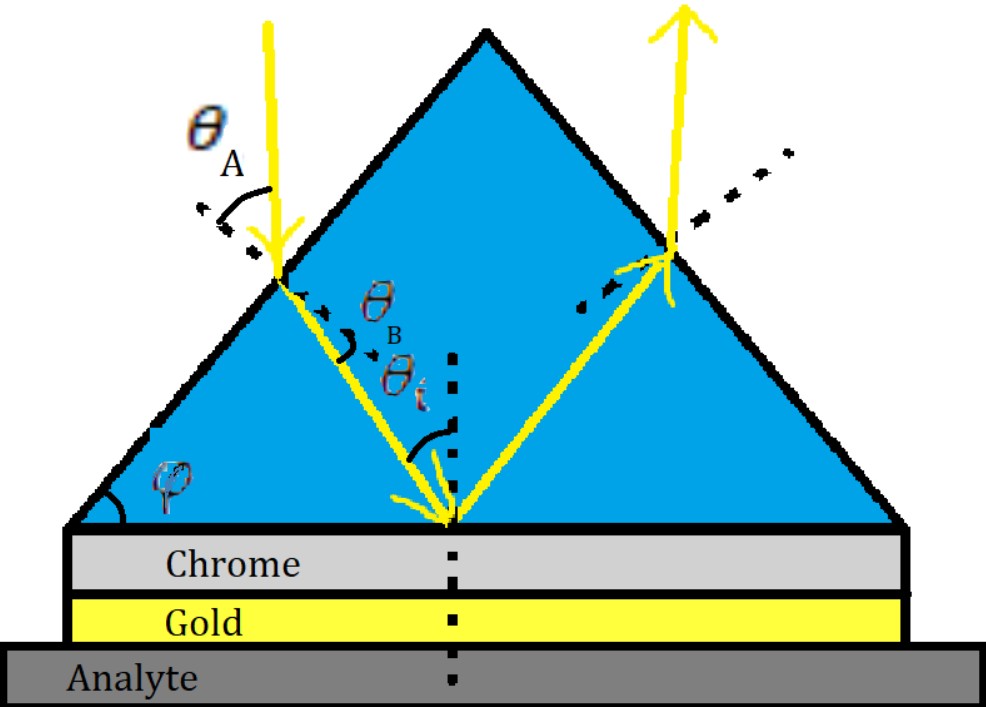

Figure 1: The Kretschmann configuration

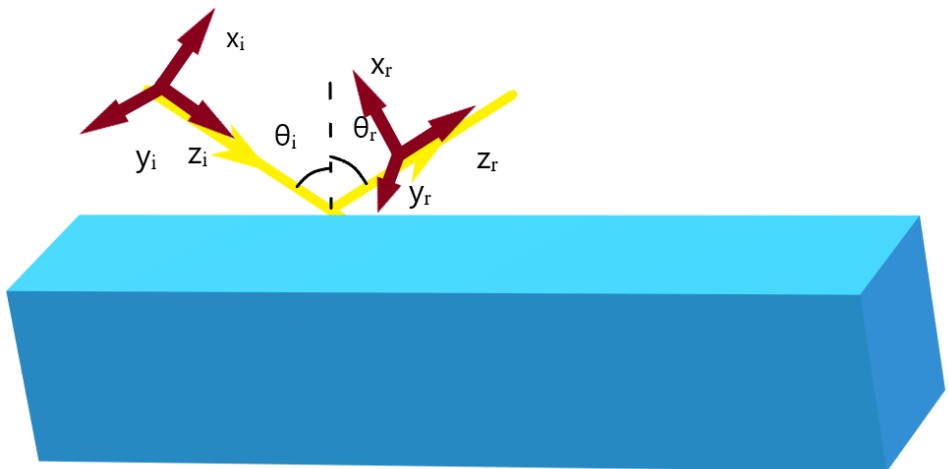

Figure 2: The geometry of the reflection of FOV with the local coordinates emphasised

## 2   Theoretical Model

In figure 2 we showcased the reflection of the LGFOV on the sandwich of metals that is presented in figure 1. Since only the transverse magnetic (TM) -polarized light can excite the plasmons [17] we will consider the reflection of an LG beam that has this polarity and with the fractional topological charge $l$. We consider, like in [18], the Snell law for refraction:

$$n_{air} \sin \theta_A = n_{prism} \sin \theta_B, \tag{1}$$

where $\theta_A$ and $\theta_B$ are the incidence angle from the air and the refraction angle from the air with $n_{air}$ and $n_{prism}$ being the indexes of refraction of air and the prism. From the geometry of the prism one can deduce the following relationship between the angle of the prism $\varphi$, the incidence angle $\theta_i$ and $\theta_B$:

$$\theta_i = \varphi - \theta_A. \tag{2}$$

For both positive and negative topological charge we can write the initial field of the LGOV (at $z = 0$) in the paraxial approximation as [19]:

$$u(x, y, z = 0) = \left( \frac{\sqrt{2}(x \pm iy)}{w_0} \right)^{|l|} e^{-\frac{x^2+y^2}{w_0^2}}, \tag{3}$$

with $+$ for $l > 0$ and $-$ for $l < 0$. For simplicity we will first consider the case when $l$ is positive. Hence, by taking the 2D Fourier transform of the initial field one can compute the angular spectrum amplitude of the incident beam [19]. So,

$$\tilde{u}(k_x, k_y) = \frac{1}{4\pi^2} \int_{-\infty}^{\infty} \int_{-\infty}^{\infty} u(x, y, z = 0) e^{-i(k_x x + k_y y)} dx dy, \tag{4}$$

where the variables $k_x$ and $k_y$ represent the transverse components of the wavevector. By Taylor expanding the function $f(x) = (1 + x)^l$ one can get an expansion similar to the binomial one but for positive fractional exponents that reads:

$$(a + b)^l = \sum_{n=0}^{\infty} a^n b^{l-n} \frac{\Gamma(l + 1)}{n! \Gamma(l - n + 1)}, \tag{5}$$

53  where $\Gamma(x)$ is the gamma function. We apply this to Eq (4) and after 2 different substi-
54  tutions we get:

$$\tilde{u}(k_x, k_y) = \int_0^\infty \int_0^{2\pi} e^{-\frac{w_0^2(k_x^2+k_y^2)}{4}-r^2} \sum_{n=0}^\infty e^{in\theta} \frac{(k_y - ik_x)^{l-n} w_0^{l-n+2} r^{n+1} \Gamma(l+1)}{\pi^2 2^{\frac{l}{2}-n+2} n! \Gamma(l-n+1)} d\theta dr. \quad (6)$$

55  By recalling the definition of the Kronecker delta function:

$$\int_0^{2\pi} e^{in\theta} = 2\pi\delta_{n,0}, \quad (7)$$

56  we have:

$$\tilde{u}(k_x, k_y) = \frac{e^{-\frac{w_0^2(k_x^2+k_y^2)}{4}} w_0^{l+2}(k_y - ik_x)^l}{4\pi\sqrt{2^l}}, \quad (8)$$

57  which is similar to the case of OVs with integer topological charges. The complex ampli-
58  tude of the incident beam can be obtained by doing the inverse Fourier transform of its
59  angular spectrum [19]. Thus, it will have the expression:

$$u_i(x_i, y_i, z_i) = \int_{-\infty}^\infty \int_{-\infty}^\infty \tilde{u}(k_x, k_y) e^{i\left(k_x x_i + k_y y_i - \frac{k_x^2+k_y^2}{2k_i} z_i\right)} dk_x dk_y, \quad (9)$$

60  where a paraxial approximation has been applied and $k_i = k_0 = \frac{2\pi}{\lambda}$. By performing similar
61  calculations as those shown before one gets:

$$u_i(x_i, y_i, z_i) = \frac{e^{-\frac{x_i^2+y_i^2}{w_0^2\left(1+\frac{iz_i}{z_{R,i}}\right)}}}{1 + \frac{iz_r}{z_{R,i}}} \left(\frac{\sqrt{2}(x_i + iy_i)}{w_0\left(1+\frac{iz_i}{z_{R,i}}\right)}\right)^l, \quad (10)$$

62  where $z_{R,i} = \frac{k_i w_0^2}{2}$ is the Rayleigth length of the incident beam.

63  Despite the fact that we considered only a TM-polarized incident beam, the reflected
64  beam will have 2 components that are TM and TE polarized. Their angular spectrums
65  can be calculated as in [19] and the results will be:

$$\tilde{u}_r^{TM}(k_{rx}, k_{ry}) = r\tilde{u}(k_{ix}, k_{iy})\left(1 - \frac{k_{rx}}{k_0}\frac{\partial r}{\partial \theta_i}\right) \quad (11)$$

$$\tilde{u}_r^{TE}(k_{rx}, k_{ry}) = -r\tilde{u}(k_{ix}, k_{iy})\frac{k_{ry}\cot\theta_i}{k_0}, \quad (12)$$

66  where $r$ represents the Fresnel reflection coefficient for TM-polarized plane waves. We also
67  impose the following conditions: $k_{rx} = -k_{ix}$ and $k_{ry} = k_{iy}$ like in [19]. By performing
68  the inverse Fourier transform of those physical quantities similar to the incident beam one
69  gets the complex amplitudes for both components of the reflected beam as follows:

$$u_r^{TM}(x_r, y_r, z_r) = ru_r\left(1 + \frac{i}{k_0}\frac{\partial r}{\partial \theta_i}\left(\frac{l(iy_r + x_r)}{x_r^2 + y_r^2} - \frac{2x_r}{w_0^2\left(1+\frac{iz_r}{z_{R,r}}\right)}\right)\right) \quad (13)$$

$$u_r^{TE}(x_r, y_r, z_r) = \frac{ru_r}{k_0(iy_r - x_r)}\left(\frac{2y_r(y_r + ix_r)}{w_0^2\left(1+\frac{iz_r}{z_{R,r}}\right)} - l\right), \quad (14)$$

where $z_{R,r} = \frac{k_r w_0^2}{2}$ is the Rayleigh length of the reflected beam, $k_r = k_0$ and:

$$u_r = \frac{\sqrt{2}^l (iy_r - x_r)^l e^{-\frac{x_r^2 + y_r^2}{w_0^2\left(1 + \frac{iz_r}{z_{R,r}}\right)}}}{w_0^l \left(1 + \frac{iz_r}{z_{R,r}}\right)^{l+1}}. \tag{15}$$

The reflectance of our optical system can be written as:

$$R = \frac{\int_{-\infty}^{\infty} \int_{-\infty}^{\infty} |u_r^{TM}(x_r, y_r, z_r)|^2 + |u_r^{TE}(x_r, y_r, z_r)|^2 dx_r dy_r}{\int_{-\infty}^{\infty} \int_{-\infty}^{\infty} |u_i(x_i, y_i, z_i)|^2 dx_i dy_i}. \tag{16}$$

The integrals in this expression are straightforward needing only a change from cartesian to polar coordinates. The final result:

$$R = |r|^2 \left(1 + \frac{l+1}{k_0^2 w_0^2} \left|\frac{\partial r}{\partial \theta_i}\right|^2 + \frac{\cot^2 \theta_i (l+1)}{w_0^2 k_0^2}\right) \tag{17}$$

is similar to the case with integer topological charge studied in [11].

For the case of negative $l$ one can do the same calculations as presented here for $l$ positive and will obtain the same expressions for the angular spectra and the complex amplitudes of the incident and the reflected beams and the reflectance but with $|l|$ instead of $l$.

# 3 Numerical Results

We consider the configuration presented in figure 1 with the thicknesses of Chrome and Gold being 10 nm and, respectively, 40 nm. For the following results who have been obtained by using a GNU Octave software, version 5.1.0, the wavelength was chosen to be $\lambda = 780 \; nm$.

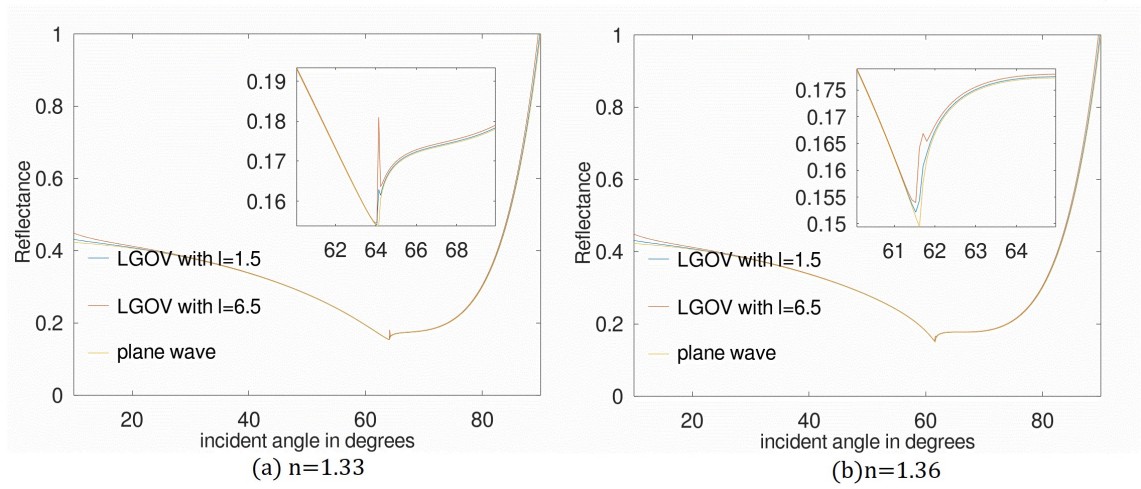

Figure 3: The reflectance in terms of the incident angle of a SPR sensor consisting of BK7 glass prism and an analyte of thickness 60 nm and a refractive index a) n=1.33 and b) n=1.36 when hit by LGOVs of OAM quantum numbers 1.5 and 6.5 and initial waist of $w_0 = 10\lambda$.

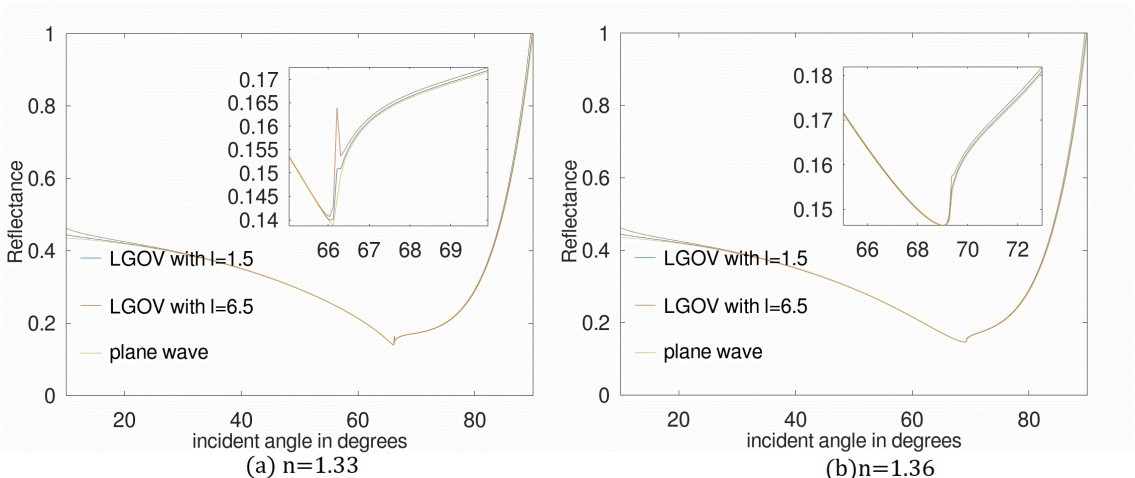

Figure 4: The reflectance in terms of the incident angle of a SPR sensor consisting of a fused-silica prism and an analyte of thickness 60 nm and a refractive index a) n=1.33 and b) n=1.36 when hit by LGOVs of OAM quantum numbers 1.5 and 6.5 and initial waist of $w_0 = 10\lambda$.

In the figures 3 and 4 one can see the plotting of the reflectance in terms of the angle $\theta_i$ for 2 different materials for the prism, namely: fused-silica and BK7 glass and 2 different values for the index of refraction for the analyte, respectively: 1.33 and 1.36. The thickness of the analyte was chosen to be 60 nm and the initial waist 10 wavelengths. Here, just like in the case of integer topological charges [11] we observe a sudden shift of its value near the resonance angle (the minimum point). That is due to the fact that we considered only the first term in the Taylor expansion of the reflection coefficient in the equations (11) and (12). Despite all that, this shift is small enough to justify this action. Also, one can notice that the bigger the index of refraction of the analyte is the smaller this shift is. Another thing to observe is the fact that the minimum point does change with the OAM quantum number but with very small values.

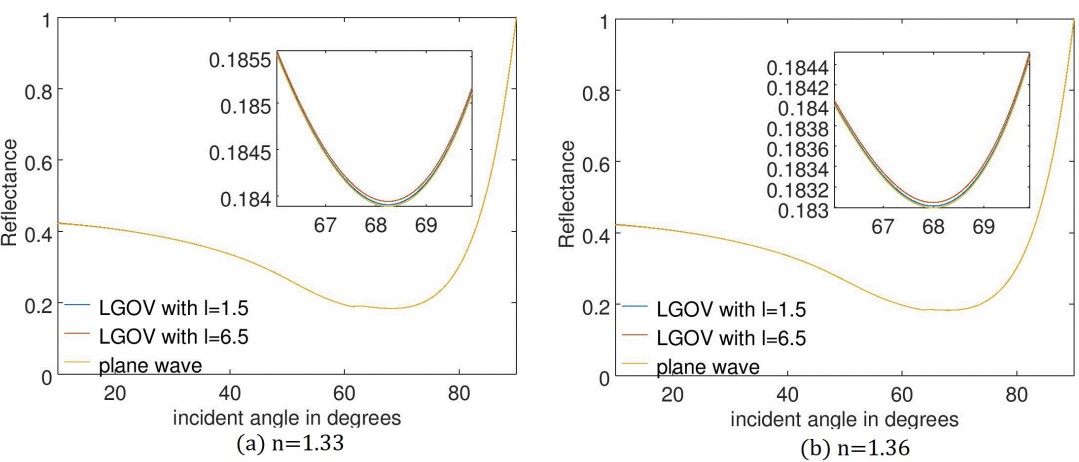

Figure 5: The reflectance in terms of the incident angle of a SPR sensor consisting of BK7 glass prism and an analyte of thickness 60 nm and a refractive index a) n=1.33 and b) n=1.36 when hit by LGOVs of OAM quantum numbers 1.5 and 6.5 and initial waist of $w_0 = 40\lambda$.

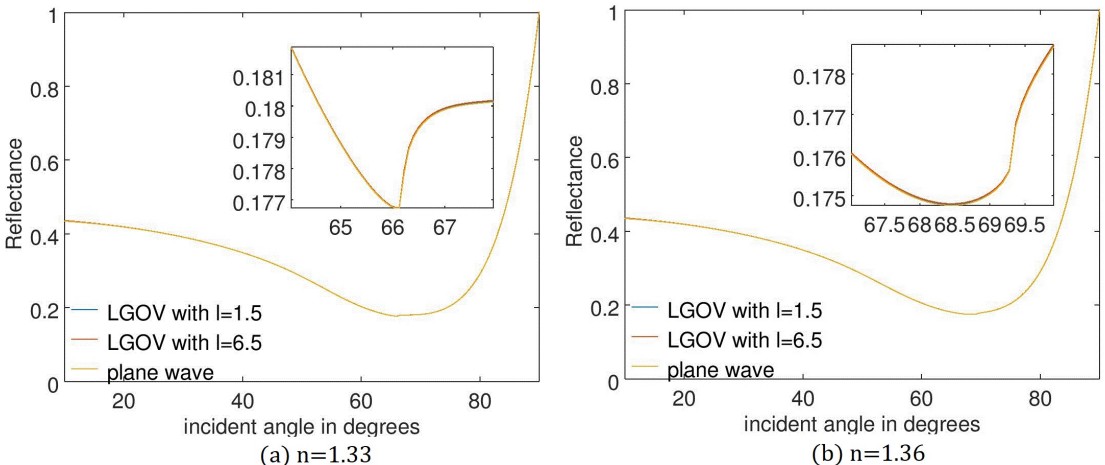

Figure 6: The reflectance in terms of the incident angle of a SPR sensor consisting of a fused-silica prism and an analyte of thickness 60 nm and a refractive index a) n=1.33 and b) n=1.36 when hit by LGOVs of OAM quantum numbers 1.5 and 6.5 and initial waist of $w_0 = 40\lambda$.

The figures 5 and 6 represent the dependance of the response in reflectance for an initial waist of 40 wavelengths and an analyte 100 nm thick. If at the previous figures one could observe a sudden shift in the response on reflectance near the resonance angle, here one cannot notice any of that. This demonstrates that the shift depends on the thickness of the analyte, its index of refraction and the OV's topological charge. Also, the resonance angle does not change at all. This means that its value depends on the thickness and the index of refraction of the analyte, on the material of the prism and weakly on the FOV's topological charge.

## 4   Conclusion

This paper has studied the response on reflectance of SPR sensors when illuminated with FOVs instead of plane waves. Similar to [11] the analytical calculus has been done by taking only the first term of the Fresnel reflection coefficient for the TM-polarized light. As a result, the analytical equation for the reflectivity is the same as the one for OVs with integer topological charges. Another similarity with [11] resides on the conclusions that could be drawn from the numerical results such as the small change of the resonance angle with the topological charge that can go unnoticed. Also, the bigger the initial waist of the FOV is the smaller the difference between it and the plane wave in reflectance is. However, the use of FOVs instead of plane waves is desirable if one needs to manipulate the particles of the analyte or the cells present in biological liquids.

## Acknowledgements

We thank the Romanian Ministry of Research, Innovation and Digitization for supporting the IMT Core Program NanoEl, within the PNCDI 2022–2026 and financing "MicroNEx", Contract nr. 20 PFE /30.12.2021.

**Funding information** This research has not received any particular external funding.

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
