# Peer review of "Surface Plasmon Resonance Sensors Using Fractional Optical Vortices"

_SciPost Physics_

## Round 1 · Referee Report · Anonymous (Referee 1) · 2025-11-29

Strengths

Detailed algebra, consistent set of results.

Weaknesses

"Negative" result, useful but not a "breakthrough" as indicated by the Author.

Report

In this manuscript, the Author calculate the reflectance of a beam from a prism in Kretschmann configuration. The peculiarity is that the profile of the beam features "fractional optical vortices" (FOVs). The Author's result, to the best of my understanding, is that the reflectivity is basically not impacted by the FOVs, and maps almost exactly that of a plane wave. This makes beams with FOVs safe to use in the Kretschmann configuration. This is certainly a noteworthy result for the experimentalist in need of using a beam with FOVs, but, from the theoretical point of view, it could appear trivial unless the Author details why a different results should or could be expected.

Requested changes

1-If possible, add a few sentences where a motivation is put forward, explaining why beams with FOVs could behave differently from plane wave in the Kretschmann configuration.

Recommendation

Ask for minor revision

  • validity: good
  • significance: ok
  • originality: low
  • clarity: ok
  • formatting: good
  • grammar: reasonable

Author:  Bulzan George Andrei  on 2025-12-04  [id 6104]

(in reply to Report 1 on 2025-11-29)
Category:
answer to question

Thank you for your report and all my appologies for the waiting!

As the equation (17) shows the reflectance is dependent on the orbital angular momentum (OAM) in the sense that 2 of the 3 terms in the paranthesis are directly proportional with l+1 where l is the OAM quatum number or the topological charge. Therefore, the lower the l value is, the lesser the FOV's impact on it is. I chose 1.5 and 6.5 as values for l in the numerical results presented but for bigger values of l bigger differences between FOV's and plane waves are expected.

As for the motivation behind this work, one can consider the FOV's capabilities to manipulate particles, to create optical traps and to sort and orientationally control cells, the last of which can lead to new applications of RPS in different domains of medicine and biology. Although, I will wait for the recomendations of the Editor-in-Charge before doing any change on the paper.

---

## Round 1 · Referee Report · Anonymous (Referee 2) · 2025-12-19

Strengths

clear presentation and discussion of results

Weaknesses

too much overlap with previous work by the same group

Report

The main problem with the present manuscript dealing with fractional optical vortices (FOV) is the strong overlap with previous work by the same group about integer optical vortices (OV), namely Ref.11. Indeed, the key analytical result given here by Eq. 17 is identical, rather than just similar, to the corresponding one for the integer case (Eq.4 in Ref.11). Also, the numerical results show a very similar behavior, probably just interpolating those of nearby integer values (a detailed comparison would have been in order). Therefore, in the absence of evidence that the use of FOV in this context is indeed superior to that of OV (and not just to that of plane waves), I cannot recommend this work for publication in SciPost.

Recommendation

Reject

---

## Editorial Decision

in_refereeing